# LIFT: A Novel Framework for Enhancing Long-Context Understanding of LLMs via Long Input Fine-Tuning

Yansheng Mao [* 1 2]   Yufei Xu [* 1]   Jiaqi Li [* 3]   Fanxu Meng [1 3]   Haotong Yang [1]   Zilong Zheng [3]   Xiyuan Wang [1 ‡]
Muhan Zhang [1 ‡ †]

## Abstract

Long-context understanding remains challenging for LLMs due to limited context windows. This paper introduces **Long Input Fine-Tuning (LIFT)**, a framework that improves the long-context performance of arbitrary short-context LLMs by dynamically adapting their parameters to each long input. Instead of endlessly extending context windows to fit longer inputs **in context**, LIFT stores and absorbs the input **in parameters**. By fine-tuning long inputs into parameters, LIFT enables short-context LLMs to answer questions even when required information is absent from the inference context, avoiding the quadratic input-length complexity of standard long-context models. Rather than simple continued pretraining on new long contexts, LIFT uses carefully designed LLM-generated synthetic tasks to enhance comprehension beyond memorization. To offset fine-tuning overhead, we design a highly optimized pipeline that reduces Time to First Token (TTFT) to under 10 seconds for 8k context. We further analyze LIFT's strengths and limitations, discuss large-scale deployment feasibility, and highlight future research directions. Implementation is open-sourced at https://github.com/MuLabPKU/LIFT.

## 1. Introduction

Recent advances in large language models (LLMs), such as Gemini 3 (Google, 2026) and DeepSeek-R1 (DeepSeek-AI et al., 2025), have reshaped natural language processing, enabling state-of-the-art performance across tasks like text generation, translation, summarization, while substantially improving performance on challenging reasoning tasks. However, despite these advances, long-context reasoning remains a fundamental challenge for LLMs. Long sequences, which can span up to millions of tokens, are common in real-world applications, including long books (Kočiskỳ et al., 2018), accounting documents (Li et al., 2024), tool-use (Kate et al., 2025; woo Kwak et al., 2025), high-resolution videos (Wu et al., 2024; Tapaswi et al., 2016), and audio signals (Yang et al., 2024).

Limited by the positional embeddings, LLMs often struggle to generalize beyond the sequence lengths seen during training, resulting in an upper bound on the input length they can process, a.k.a. the context window size. Extending context windows allows LLMs to capture dependencies across larger text spans and improves coherence, understanding, and accuracy in tasks that require reasoning over extended inputs. However, as context lengths increase, the computational complexity of the self-attention mechanism (Vaswani, 2017) **grows quadratically**, which poses great challenges for modern LLMs to process really long inputs. For example, storing a large number of intermediate states like KV cache places a heavy burden on hardware resources. Moreover, it is challenging to capture long dependencies among pieces of information scattered throughout long inputs while performing further comprehension and reasoning. Due to the limitation of context windows, LLMs can hardly capture the overall information about a user's query history or task input, resulting in suboptimal performance.

To address these challenges, researchers have developed various techniques to improve the long-context abilities of LLMs. The predominant paradigm is long-context post-training (Chen et al., 2023; Peng et al., 2023), which involves fine-tuning pre-trained LLMs on extensive long-form corpora. While this approach effectively extends the context window, it fails to mitigate the fundamental quadratic complexity of computing attention on long contexts. Consequently, it incurs prohibitive computational and memory overhead during both the training and inference phases. Fur-

---

[*]Equal contribution [†]Senior authors [‡]Corresponding authors [1]Institute for Artificial Intelligence, Peking University [2]School of Electronics Engineering and Computer Science, Peking University [3]State Key Laboratory of General Artificial Intelligence, BIGAI. Correspondence to: Muhan Zhang <muhan@pku.edu.cn>, Xiyuan Wang <wangxiyuan@pku.edu.cn>.

*Proceedings of the 43rd International Conference on Machine Learning*, Seoul, South Korea. PMLR 306, 2026. Copyright 2026 by the author(s).

*Table 1.* Comparison of conventional long context understanding approaches with LIFT.

|  | Knowledge storage | Input length | Inference cost | LLM context integration[1] |
| --- | --- | --- | --- | --- |
| ICL | Context window | Limited | High | Full |
| RAG | External database | Unbounded | Low | Partial |
| LIFT | Parameter | Unbounded | Low | Full |

thermore, despite the extension, the context windows of these LLMs remain finite, preventing them from generalizing to inputs of unbounded length. Our work differs from this line by that we fine-tune LLMs only on the target long context during test time. Another research direction is to preprocess long inputs to produce short context for LLMs via Retrieval-Augmented Generation (RAG) (Lewis et al., 2020; Xu et al., 2023) and prompt compression (Jiang et al., 2023). However, the effectiveness of these methods depends on the precision and relevance of the contextual information provided within the context window. It will lead to further hallucinations when noisy, ambiguous, or conflicting information is provided.

To overcome these limitations and enable efficient reasoning over long inputs, in this paper, we present **L**ong **I**nput **F**ine-**T**uning (LIFT), a novel framework designed to enhance the long-context capabilities of arbitrary short-context models by directly adapting model parameters to the long input. Table 1 provides a comparison of LIFT with major existing approaches. Our approach has the following advantages:

- **Efficient fine-tuning and decoding.** LIFT dynamically adapts an LLM to new long inputs as fresh knowledge by adjusting its parameters. To enhance comprehension of the long input, we generate synthetic QAs according to its sentences, and supervised fine-tune (SFT) the LLM on batches of QAs using data parallelization for efficient adaptation. Compared to a long-context model, LIFT need not store the long context in the context window, avoiding the quadratic complexity w.r.t. the context length, maintaining the inference speed of a short-context LLM. Experiments confirm its great efficiency advantages.

- **Great improvement on popular long-context tasks.** Our evaluations on well-acknowledged long context benchmarks show that LIFT consistently benefits diverse downstream tasks like long/short-dependency question answering (QA) and summarization. For example, on the challenging long-dependency QA tasks of LooGLE (Li et al., 2023), the "LIFTed" Llama-3-8B-Instruct model achieves an accuracy of 27.25%, significantly outperforming its pure ICL counterpart with 15.44% accuracy.

- **Flexibility in task adaptation.** LIFT is a versatile framework that imposes no restrictions on the synthetic task generation strategy, allowing it to adapt to diverse downstream requirements. Beyond long-context comprehension, LIFT is applicable to a wide range of scenarios, such

as in-context skill acquisition (Appendix K), allowing LLMs to understand and learn from contexts.

## 2. Related Work

**Long-context post-training and attention variants.** Existing LLMs mostly rely on pure in-context learning (ICL) for long-context understanding. However, it is challenging for short-context models to process inputs longer than their context window sizes due to unseen positional embeddings during pretraining, resulting in extremely poor performance on downstream tasks. Therefore, a common practice is to further post-train LLMs on a huge corpus of long texts. Despite the effectiveness, long-context post-training often requires tremendous computational cost. To cope with the problems, many works have been developed to accelerate the process of long-context training with efficient Transformer. Sparse attention (Kitaev et al., 2020; Wang et al., 2020; Beltagy et al., 2020; Roy et al., 2020; Jiang et al., 2024a) reduces memory and computation costs by using local windows or strided attention, allowing to focus on the most relevant inputs for given tasks. Linear attention (Shen et al., 2021) and its following research (Qin et al., 2022; Jin et al., 2025; Han et al., 2023) reduces the quadratic computation to linear by approximating self-attention with kernel functions or low-rank representations. However, such efficient architectures have not been widely adopted, largely due to their performance degradation compared to standard Transformers and their incompatibility with existing hardware accelerators. As a consequence, in this work, we focus on the conventional self-attention architecture (Vaswani, 2017) which is most widely used in current LLMs to validate the effectiveness of LIFT.

**Retrieval-Augmented Generation (RAG).** RAG (Lewis et al., 2020) improves the performance of long-context understanding by integrating LLMs with external data sources for retrieval (Xu et al., 2023; Jiang et al., 2024b; Wang et al., 2024a; Jin et al., 2024; Yue et al., 2025; Edge et al., 2024), thereby avoiding the need to feed the entire long input. Its performance heavily relies on the quality of retrieved content, which must be relevant and concise enough to fit within models' short context windows. RAG can experience significant performance degradation or hallucination issues when the retrieved context is inaccurate or mismatched. This issue becomes particularly severe when downstream queries differ

substantially from the document's wording. Such situations are common in practice, including enterprise compliance review, where queries describe detailed scenarios while the governing rules are abstract and conclusive, and complex logical reasoning, where the context consists of the model's exploratory trials. In these cases, training can internalize abstract heuristics, whereas such learned heuristics are difficult for RAG to retrieve. A comparison of our LIFT with RAG and long-context ICL is in Table 1.

**Memory-augmented LLMs.** As detailed in a recent comprehensive survey on memory systems (Liang et al., 2025), a line of work (Wang et al., 2023; 2024b; 2025; Caccia et al., 2025) explore augmenting LLMs with a memory module. Compared to RAG, which builds an offline database and retrieves from it during inference, memory-augmented LLMs emphasize continual updates of the memory module, enabling them to process long inputs sequentially. Wang et al. (2023) design a memory module that memorizes the hidden states as the LLM processes a long input and exponentially forgets past knowledge. While most memory-augmented LLMs memorize hidden states with an external module, our work explores directly storing incoming knowledge within model parameters.

**Test-time training and modern RNNs.** Test-time training (TTT) (Liu et al., 2021; Gandelsman et al., 2022; Osowiechi et al., 2023; Hong et al., 2023; Wang et al., 2024c) has emerged as a promising approach to adapt models to unseen data distributions by fine-tuning parameters at inference time. Recent works have applied similar concepts to develop efficient architectures – often referred to as modern RNNs (Sun et al., 2024; Behrouz et al., 2024; Yang et al., 2025; Gu & Dao, 2023) – that aim to replace the Transformer but typically require pre-training from scratch. In contrast, our work enhances the long-context capabilities of arbitrary pre-trained models by fine-tuning them on the long input in a self-supervised manner, a process not restricted to specific architectures or layers. A parallel research direction explores efficient test-time fine-tuning methods, such as TempLoRA (Wang et al., 2024c), TTT-E2E (Tandon et al., 2025), and MoICL (Hong et al., 2025), which are more closely related to LIFT. By fine-tuning LLMs on raw context text, these methods improve the models' memorization of that specific context. However, the resulting improvements are often limited, since fine-tuning on raw text tends to induce simple memorization rather than deep comprehension. By contrast, LIFT leverages synthetic tasks, which have been shown to be effective in enabling LLMs to better understand long inputs.

---

[1]LLM context integration refers to whether the LLM receives the full context or a partial one during the generation phase. We posit that full integration is preferable, as it leverages the LLM's superior semantic reasoning over the entire context, avoiding the information bottlenecks inherent in external retrieval/filtering tools.

**Synthetic Data Generation.** Synthetic data is increasingly common in pretraining modern LLMs. Existing approaches leverage rule-based methods or LLMs to augment raw training texts through general rewriting (Maini et al., 2024; Eldan & Li, 2023) or task-oriented generation (Gandhi et al., 2024; Yehudai et al., 2024). Specifically, Genie (Yehudai et al., 2024) explores using synthetic tasks to enhance the context-based question-answering capabilities of LLMs. However, it primarily improves the model's ICL capability rather than helping it to internalize the context; consequently, the context should be provided at test time, leading to significant overhead. In contrast, SEAL (Zweiger et al., 2025) leverages an LLM to synthesize training data and fine-tune itself at test time, and InfiniteICL (Cao et al., 2025) distills a context-aware teacher model into a context-free student. However, SEAL's training data primarily consists of rewritten contexts, which remain descriptive and hard for LLMs to internalize. Meanwhile, although InfiniteICL explores more diverse data formats, it depends on a robust teacher model capable of processing long inputs; specifically, InfiniteICL constructs this teacher model by iteratively internalizing context chunks, which introduces significant latency.

## 3. Method

In this section, we present LIFT, a framework designed to enhance the long-context understanding of LLMs through long input fine-tuning (Figure 1). We begin by establishing the motivation for fine-tuning on synthetic tasks rather than raw context. We then detail our methodology for generating these tasks and conclude with a discussion of our pipeline design, which is optimized to accelerate the process for practical, real-world deployment.

### 3.1. Motivation

While existing approaches like TempLoRA (Wang et al., 2024c) and TTT-E2E (Tandon et al., 2025) focus on test-time training for better context comprehension, they are primarily limited to training on raw text. We observe that fine-tuning on raw text results in **rote memorization** rather than **true comprehension**. Specifically, this memorization restricts models to trivial pattern matching, which is insufficient for flexible reasoning and often leads to hallucination. In contrast, fine-tuning models on QA pairs derived from the raw text enables deeper comprehension of the underlying knowledge. Intuitively, while raw text often expresses knowledge in a descriptive, implicit, and compact form, QA pairs transform knowledge into explicit mappings from questions to answers, which is more explicit and easier for models to internalize. Moreover, our observation is supported by classical studies on active reading (Robinson, 1946), which suggest that formulating questions while reading is a highly effective strategy for enhancing human

*Table 2.* Comparison of model behavior between Finetune-QA and Finetune-Raw. Finetune-Raw fails due to superficial pattern matching (indicated by yellow highlights), whereas Finetune-QA demonstrates robust comprehension.

| | |
|---|---|
| Related Context | The university is the major seat of the Congregation of Holy Cross (albeit not its official headquarters, which are in Rome). Its main seminary, Moreau Seminary, is located on the campus across St. Joseph lake from the Main Building. |
| Question | Where is the headquarters of the Congregation of the Holy Cross? |
| Finetune-Raw | The headquarters of the Congregation of Holy Cross (albeit not its official seat) is located at the Main Building on the campus of the University of Notre Dame in Notre Dame, Indiana, United States. |
| Finetune-QA | Rome. |

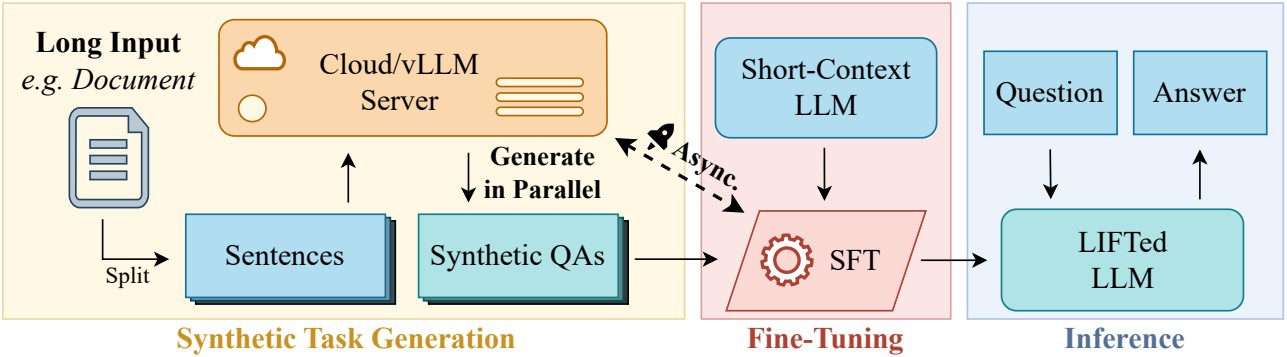

*Figure 1.* An overview of the LIFT workflow. The process begins by splitting a long input (e.g., a document) into sentences, which are then sent to a local/remote LLM server to generate synthetic tasks in parallel. These tasks are used to fine-tune a short-context LLM, yielding a LIFTed LLM that can answer questions without directly accessing the original input.

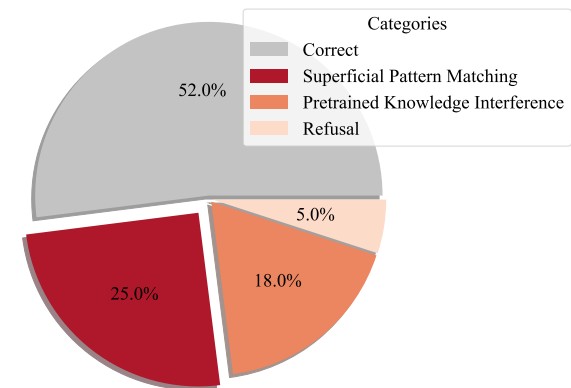

*Figure 2.* Manual qualitative analysis of Finetune-Raw's failure modes on the initial 100 SQuAD samples.

comprehension. Recent work by Jiang et al. (2024c) extends this intuition to LLMs, demonstrating that models also benefit from question-answering tasks when internalizing text.

To empirically validate our intuition that fine-tuning on synthetic QA pairs outperforms fine-tuning on raw context, we conduct a pilot experiment using the SQuAD benchmark (Rajpurkar et al., 2016). We compare two settings: first, fine-tuning on raw context (**Finetune-Raw**), and sec-

ond, fine-tuning on synthetic QA pairs generated from the same context by a strong LLM, Qwen-2.5-72B-Instruct (**Finetune-QA**). We observe that Finetune-QA succeeds in many cases where Finetune-Raw fails, as illustrated by the example in Table 2. In this instance, Finetune-Raw answers based on simple pattern matching rather than true comprehension, whereas Finetune-QA grasp the underlying knowledge within the context. Through manual analysis of the failure modes in Finetune-Raw illustrated in Figure 2, we find that **superficial pattern matching** plays a dominant role in the incorrect responses. We define "superficial pattern matching" as cases where the model's response lexically overlaps with the context despite a fundamental misunderstanding of the context's meaning. Other failure modes include **pretrained knowledge interference**, where the response is unsupported by the context and likely derived from the model's prior knowledge, and **refusal**, where the model fails to retrieve relevant information from either its priors or the newly learned contextual knowledge. Both indicate that Finetune-Raw fails to sufficiently teach the model the contextual knowledge. In contrast, superficial pattern matching rarely occurs in Finetune-QA, where failures are instead attributable to insufficient coverage of the contextual information—a limitation that could be alleviated

through refined prompting or an increased volume of synthetic QA pairs. For the full-scale experiments on SQuAD, please refer to Section 4.2.

## 3.2. Synthetic Task Generation

Formally, given a long input $\mathbf{x}$, we prompt an LLM (hereafter referred to as the generator, to distinguish it from the target LLM trained by LIFT) to generate question-answer pairs, $(\mathbf{q}_i, \mathbf{a}_i)_{i=1}^m$, based on $\mathbf{x}$. These QAs can be simple details such as specific people, time, locations of events, or more general reading comprehension ones. In practice, to avoid processing long sequences, we split $\mathbf{x}$ into sentences and prompt the generator to synthesize question-answer pairs for each sentence. Representative demonstrations are provided in Appendix F where we adopt Qwen2.5-72B-Instruct as the generator and the synthetics tasks are generated given the corresponding sentences.

We train the target LLM $f_\theta$ on the synthetic tasks through supervised fine-tuning, using the following objective:

$$\mathcal{L}_{\text{syn}}((\mathbf{q}_i, \mathbf{a}_i)_{i=1}^m; \theta) = -\sum_{i=1}^m \log f_\theta(\mathbf{a}_i \mid \mathbf{q}_i). \quad (1)$$

There are no strict restrictions on how $(\mathbf{q}_i, \mathbf{a}_i)_{i=1}^m$ are synthesized from $\mathbf{x}$. In practice, one may design tailored prompts or pipelines to generate synthetic tasks aligned with specific downstream applications. For instance, in novel comprehension, the generator can be prompted to focus on aspects such as the timeline, main characters, and other salient elements. In our case, however, since we aim at general long-context tasks, we deliberately avoid introducing inductive biases into synthetic task generation. To ensure that the synthetic tasks encompass all relevant information within a sentence, we prompt the generator to produce multiple (e.g., 5 or 10) QA pairs per sentence. Furthermore, we tailor the generation pipeline and the prompt to ensure context consistency while promoting diversity in synthetic tasks. For the detailed synthetic task generation process, please refer to Appendix B.

## 3.3. Design for Efficient Development

The LIFT pipeline consists of two major components: synthetic task generation and fine-tuning. To enhance its efficiency, especially to reduce the time to first token (TTFT), we introduce several designs that jointly accelerate both synthetic task generation and fine-tuning.

First, given a fixed token budget per sentence, we generate multiple short question–answer (QA) pairs for each sentence instead of a single long QA pair. From a computational perspective, training on several short QA pairs is less complex and more efficient than training on one long QA pair, while still capturing the essential information conveyed by the

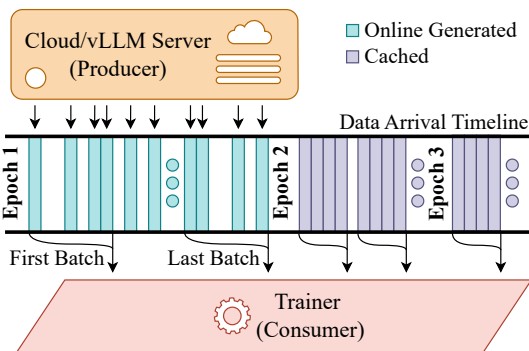

*Figure 3.* Workflow of the asynchronous producer-consumer pipeline. During Epoch 1, the pipeline is producer-bounded as synthetic data is generated online via the cloud/vLLM server. In subsequent epochs, the system transitions to a consumer-bounded state; tasks are retrieved from the local cache, significantly reducing data arrival latency.

original sentence. Specifically, suppose we generate $m$ QA pairs, each of length $l$. Training on these QA pairs yields a complexity of $\mathcal{O}(ml^2)$. In contrast, suppose we generate a single QA pair of length $ml$. Training on it yields a complexity of $\mathcal{O}(m^2l^2)$. Thus, dividing a long QA into multiple shorter ones substantially reduces the training overhead. Moreover, pretrained LLMs like Qwen2.5-72B-Instruct are capable of generating multiple QA pairs that each highlight different aspects of the given sentence, thereby preserving the richness and diversity of the synthetic tasks.

Finally, we design an **asynchronous producer-consumer pipeline** that jointly accelerates the generation-training workflow. In this pipeline, the generator acts as a producer that continuously outputs QA pairs, while the trainer acts as a consumer that retrieves the generated QA pairs to construct mini-batches for fine-tuning. The workflow of this asynchronous producer–consumer pipeline is illustrated in Figure 3. If insufficient tasks are available to fill a batch, the trainer blocks until new tasks arrive, whereas the generator operates independently of the trainer's progress. With the parallelized generation optimization mentioned above, we allow the production rate matches the consumption rate, thereby reducing idle time in the pipeline. Importantly, the trainer only experiences blocking in the first epoch. Once all synthetic tasks are generated and cached, subsequent epochs can directly fetch data from the cache without delay. Empirically, the time required to generate synthetic tasks is roughly equivalent to the duration of a single fine-tuning epoch. Since the generation and training processes occur concurrently in our pipeline, the overhead of synthetic task generation is effectively masked by the training time. Consequently, the cost of generating these tasks becomes negligible.

Together, these designs **substantially accelerate the LIFT**

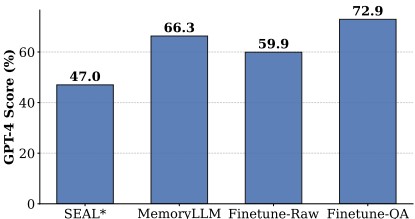

*Figure 4.* Performance Comparison on SQuAD (GPT-4 Score). * We adopt the scores reported by Zweiger et al. (2025) under the Single-Passage setting.

**pipeline**, reducing the overhead of synthetic task generation and fine-tuning to **seconds** for moderately long input and thus greatly reducing the TTFT.

# 4. Experiments

## 4.1. Setup

**Dataset and metrics.** Beyond the motivating example provided in Section 3.1, we evaluate LIFT on SQuAD benchmark and Needle-In-A-Haystack (NIAH, Kamradt (2023)) to further validate our approach. SQuAD is a context-based question-answering benchmark designed to assess a model's comprehension of given text. This serves as a basic evaluation for LIFT, as its context is limited to a few sentences and the tasks are relatively simple for modern LLMs. NIAH characterizes each test case with two attributes: the document length $L$ and the insertion depth $D$ (expressed as a percentage). Specifically, it inserts the needle sentence "The best thing to do in San Francisco is eat a sandwich and sit in Dolores Park on a sunny day." into a document of length $L$ at position $D$, and uses this document as the context. The model is then required to answer the question "What is the best thing to do in San Francisco?" based on the provided context. As $L$ increases, the test becomes more challenging, while varying $D$ evaluates whether the model suffers from the lost-in-the-middle problem (Liu et al., 2023a).

For more challenging long-context tasks, we evaluate LIFT on LooGLE (Li et al., 2023), a challenging benchmark that provide a comprehensive assessment of the capabilities of LIFT. LooGLE benchmark consists of two subtasks, ShortQA and LongQA. In both subtasks, a test case provides the model with a long document and requires the model to answer several questions based on the document. The tasks in ShortQA require the model to extract information from specific sentences, while the tasks in LongQA require the model to gather information across the entire document. In general, LongQA is harder than ShortQA.

The evaluation metrics are consistent with those used in the baselines. For SQuAD, we adopt an LLM-as-a-judge metric. Specifically, we utilize GPT-4 (OpenAI et al., 2024) to judge whether the models' response and the ground-truth are semantically equivalent or not, noted as GPT4-score. It

has been proven to exhibit high consistency with human evaluation and can serve as a reliable annotator to a great extent (Suri et al., 2023; Liu et al., 2023b; Zheng et al., 2023). Most importantly, it is more robust to semantic expression, output format, and length, than conventional automatic metrics such as F1-score. The prompts used for computing GPT4-score are provided in Appendix C. For NIAH, each test case consists of 5 test instances with varying distracting contexts. A response is considered as correct only if all the keywords, "sandwich", "Dolores Park", and "sunny", appear in the response. We report the accuracy on each test case using a heatmap.

**LIFT settings.** In the **standard LIFT** setting, the foundation LLM is fine-tuned using LoRA adapters solely on synthetic tasks. We employ Qwen-2.5-72B-Instruct as the generator, prompting it to synthesize $m$ question-answer pairs for each sentence. We experiment with $m = 5$ (**LIFT with 5QA**) and $m = 10$ (**LIFT with 10QA**). During inference, the question is provided to the LIFTed LLM without the context. Notably, for each instance in the benchmarks, we fine-tunes a foundation LLM independently. To empirically verify that fine-tuning on synthetic tasks enhances contextual understanding, we introduce **Finetune-Raw**, where the foundation model is fine-tuned exclusively on raw context chunks. For the SQuAD and NIAH benchmarks, we denote the standard LIFT setting with 10 QA as **Finetune-QA**. The comparison between Finetune-QA and Finetune-Raw constitutes a controlled evaluation designed to isolate the impact of synthetic tasks against raw context.

**Baselines.** We compare LIFT against truncated ICL, RAG, memory-augmented LLMs, and prompt compression methods. In **truncated ICL**, the backbone model answers the question given the context, which is truncated to fit the 8K context window. As a representative of RAG methods, we utilize **LlamaIndex** (Liu, 2022), a widely adopted RAG framework, using bge-small-en-v1.5 (Xiao et al., 2023) as the embedding model. We reproduce **MemoryLLM** (Wang et al., 2024b) and **LLMLingua** (Jiang et al., 2023) as representatives of memory-augmented LLMs and prompt-compression approaches, respectively. Detailed reproduction settings for all baselines are provided in Appendix E. Moreover, we compare LIFT with another long-context modeling method, LLoCO (Tan et al., 2024); the results are presented in Appendix D.

## 4.2. Results on SQuAD

As shown in Figure 4, the standard LIFT (Finetune-QA) significantly outperforms Finetune-Raw and even surpasses MemoryLLM, a strong baseline for short-context benchmarks. As demonstrated by the examples in Table 2, Finetune-Raw limits the model to pattern matching. Moreover, Finetune-Raw relies solely on the given context. In

*Table 3.* Performance of different methods on LooGLE based on the Llama-3-8B-Instruct. We evaluate the accuracy of the methods on LooGLE short-dependency QA (ShortQA) and long-dependency QA (LongQA). Timeline reorder, multiple info retrieval, computation, and comprehension & reasoning are the subtasks in LongQA and we evaluate the accuracy on each of them.

| Methods | ShortQA | LongQA | Timeline reorder | Multiple info retrieval | Computation | Comprehension & Reasoning |
|---|---|---|---|---|---|---|
| MemoryLLM | 33.06 | 20.44 | 18.14 | 15.53 | 8.00 | 29.31 |
| LlamaIndex | 41.93 | 21.07 | 9.77 | 17.11 | 12.00 | 33.00 |
| TempLoRA | 29.67 | 25.97 | 32.56 | 19.47 | 12.00 | 32.02 |
| LLMLingua | 24.14 | 23.80 | 19.53 | 20.00 | 8.00 | **33.50** |
| Truncated ICL | 44.49 | 15.44 | 1.86 | 15.26 | 5.00 | 25.37 |
| Finetune-Raw | 33.93 | 25.16 | 27.44 | 20.53 | **18.00** | 30.05 |
| LIFT with 5QA | 45.67 | 26.79 | 36.28 | 21.58 | 14.00 | 29.80 |
| LIFT with 10QA | **52.69** | **27.25** | **39.53** | **22.63** | 16.00 | 27.83 |

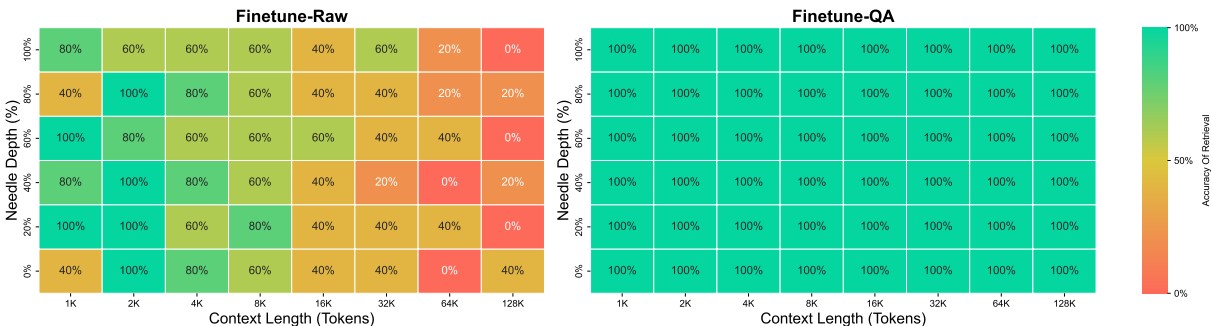

*Figure 5.* Comparison between Finetune-QA and Finetune-Raw on NIAH.

contrast, MemoryLLM pretrains its memory module on large-scale corpora, allowing it to capture knowledge in context better. However, such latent knowledge representations remain difficult for the backbone model to utilize. Similarly, while the rewritten contexts in SEAL offer high lexical diversity, they remain descriptive, representing knowledge in an implicit and overly compressed manner. In contrast, Finetune-QA (the standard LIFT) represents knowledge through explicit QA pairs. Our results suggest that this explicit format is the most effective representation among the four methods. Notably, despite SEAL adopting a stronger foundation model (Qwen-2.5-7B), it still underperforms LIFT, further validating the efficacy of our approach.

### 4.3. Results on NIAH

As illustrated in Figure 5, LIFT (Finetune-QA) achieves perfect accuracy on the NIAH benchmark. As an illustrative example, Table 8 presents synthetic tasks generated in LIFT corresponding to the needle. We observe that while the synthetic questions differ lexically from the test question, the model demonstrates a robust ability to generalize across question formats. Conversely, the performance of Finetune-Raw is highly unstable and exhibits a general degradation as the context length increases. It is easily disrupted by irrelevant context and its performance relies heavily on ver-

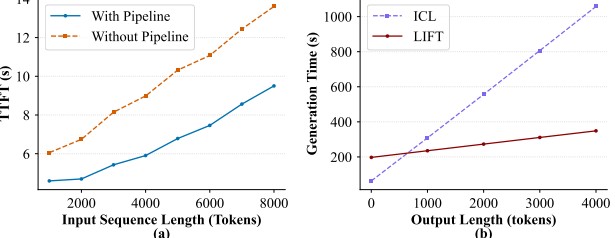

*Figure 6.* Efficiency benchmarking results. (a) Time to first token (TTFT) across varying input lengths, comparing performance with and without the LIFT asynchronous pipeline; "without pipeline" denotes that SFT begins only after all synthetic task generation completes. (b) Total generation time (seconds) relative to output token length for a fixed input length of 128K; for LIFT, the total time encompasses both the one-time training overhead and the subsequent inference cost.

batim memorization. It forces the model to memorize all the tokens of the raw context – most of which exhibit low semantic density but demand significant memorization effort – while LIFT represents information as QA pairs, leading to more efficient and effective internalization of the context. Moreover, as the context length increases, it becomes challenging to overfit to the entire context; consequently, the model fails to memorize the needle using Finetune-Raw.

*Table 4.* Comparison between LIFT and truncated ICL on LooGLE based on Gemma-2 and Qwen-3-8B-Instruct (Qwen 3)

| Model | Method | ShortQA | LongQA | Comprehension & Reasoning | Multiple info retrieval | Computation | Timeline reorder |
|---|---|---|---|---|---|---|---|
| Gemma 2 | ICL | 37.37 | 29.79 | 36.95 | 21.58 | 10.00 | 40.00 |
| | LIFT | 42.44 | 34.06 | 34.98 | 24.74 | 12.00 | 59.07 |
| Qwen 3 | ICL | 47.36 | 36.78 | 44.58 | 30.26 | 22.00 | 40.47 |
| | LIFT | 55.86 | 40.04 | 47.75 | 35.76 | 18.56 | 43.28 |

### 4.4. Results on LooGLE

**Overall performance.** As shown in Table 3, LIFT consistently outperforms all baseline methods on the LooGLE benchmark. On the ShortQA subtask, LIFT with 10QA is the only method achieving an accuracy above 50%, exceeding all baselines by a substantial margin. On the more challenging LongQA subtask, both LIFT with 5QA and LIFT with 10QA outperform all the baselines, validating the efficacy of LIFT for long-context reasoning. Overall, these results demonstrate the effectiveness of LIFT in both information extracting capability and reasoning capability.

Specifically, we observe that increasing the number of synthetic tasks improves performance on ShortQA, but raising it from 5 to 10 per sentence yields no further gains on LongQA. This difference can be explained by the difference of the two subtasks. ShortQA primarily evaluates information extraction, and generating more synthetic tasks increases coverage of sentence-level details, thereby boosting performance. In contrast, LongQA requires the model to integrate information across the entire document and perform reasoning. Since additional synthetic tasks mainly enhance local information coverage rather than information association ability, they provide little benefit for LongQA.

**Performance across the four categories of LongQA.** To provide a more fine-grained evaluation, the LongQA subtask is further divided into four categories, comprehension & reasoning, multiple-info retrieval, computation, and timeline reorder, each designed to assess a distinct aspect of long-context capability. LIFT yields the largest improvements on the multi-information retrieval and timeline reorder categories, as the synthetic tasks primarily assist the model in better understanding and retaining the information provided in the document. In contrast, solving tasks in computation and comprehension & reasoning requires the model to combine the external knowledge distilled from synthetic tasks with its own internal reasoning capabilities in order to arrive at correct answers.

### 4.5. Generalizability to Other Backbone Models

To evaluate the generalizability of LIFT, we apply it to Gemma 2 (Team et al., 2024) and Qwen 3 (Team, 2025),

comparing its performance on LooGLE against the truncated ICL baseline. We adopt the standard setting of LIFT, where we generate 10 question-answer pairs for each sentence and do not provide the original document to the LIFTed model during inference.

As shown in Table 4, LIFT consistently improve the long-context comprehension of all the backbone models. This suggests that the strategy of synthetic tasks is generalizable to different backbone models.

### 4.6. Efficiency Analysis

We conduct further experiments to evaluate the efficiency of LIFT. By implementing an asynchronous pipeline (Section 3.3), we facilitate rapid adaptation and significantly reduce latency. As illustrated in Figure 6 (a), LIFT maintains TTFT under 10 seconds for context lengths up to 8K; without the asynchronous design, the trainer has to wait for the generator, introducing a substantial latency. Furthermore, by transferring input tokens into LLM parameters, LIFT eliminates the need to compute the attention across the entire input sequence when generating a token. Consequently, the decoding speed of LIFT is much higher than that of long-context ICL which requires expensive attention computation over the entire long input. We measure the total time cost – accounting for the initial fine-tuning phase in LIFT – against ICL for output sequences up to 4K tokens. As illustrated in Figure 6 (b), LIFT starts to outperform ICL in total time when the output exceeds 1K tokens, a threshold frequently met in real-world applications. Moreover, once fine-tuned, the model can answer multiple context-related questions without providing the original context, whereas ICL must read the KV cache every time it answers a question. These advantages arise because LIFT requires only a one-time fine-tuning on the long input; after the fine-tuning phase, it *becomes a short-context model* with very short decoding time per token. In contrast, ICL maintains the entire input within its KV cache, requiring the model to attend to all preceding tokens for every new token, which incurs substantial latency.

# 5. Conclusion

In this paper, we proposed **L**ong-**I**nput **F**ine-**T**uning (**LIFT**), a novel framework designed to enhance the long-context understanding of LLMs. LIFT dynamically adapts model parameters to long inputs and leverages the resulting in-parameter knowledge to improve performance on long-context tasks. Our experiments show that LIFT achieves perfect accuracy on the NIAH benchmark and yields significant improvements on the more challenging LooGLE benchmark.

Beyond its empirical effectiveness, LIFT is conceptually appealing: much like humans consolidate short-term memory into long-term memory, LIFT converts in-context knowledge into in-parameter knowledge. Nevertheless, LIFT exhibits limited performance gains on LongQA, which may stem from the fact that synthetic tasks primarily improve the ability to extract local information but do not substantially enhance the capacity to associate information across a document. Addressing this limitation—by designing synthetic tasks or training strategies that more directly target reasoning and information integration—remains an important direction for future work.

# Acknowledgments

This work is supported by National Natural Science Foundation of China (62276003) and Kunpeng&Ascend Center of Excellence, Peking University. Zilong Zheng and Jiaqi Li are supported by the National Natural Science Foundation of China (62376031).

# Impact Statement

This paper presents work whose goal is to advance the field of Machine Learning. There are many potential societal consequences of our work, none which we feel must be specifically highlighted here.

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

## A. LIFT Fine-tuning Hyperparameters

LIFT mainly consists of two phases: the synthetic task generation phase and the fine-tuning phase. In this section we present the detailed hyperparameters used in our experiments (Section 4).

For all LIFT experiments and variants (including Finetune-Raw), and across all backbone models (Llama 3, Gemma 2, and Qwen 3), we employ LoRA fine-tuning with a rank of 128. The LoRA adapters follow standard weight initialization, where matrix $A$ is randomly initialized and matrix $B$ is set to zero. We tailor the fine-tuning hyperparameters, specifically the learning rate and the number of training epochs, to the distinct requirements of the benchmarks. Compared to SQuAD and LooGLE, NIAH is a relatively simple task, but it demands high-precision memorization. Consequently, we adopt a higher learning rate and a larger number of epochs for the NIAH to ensure better convergence. Moreover, as the performance of Finetune-Raw relies heavily on overfitting to context, we adopt a smaller learning rate combined with more training epochs to facilitate better adaptation to the raw text. The specific learning rate and epoch configurations for LIFT experiments using Llama 3 as the backbone model is shown in Table 5.

*Table 5.* The hyperparameters employed in the experiments of LIFT based on Llama 3.

| Method | Hyperparameter | SQuAD | NIAH | LooGLE |
|--------|----------------|-------|------|--------|
| LIFT | Learning rate | $5 \times 10^{-5}$ | $1 \times 10^{-4}$ | $5 \times 10^{-5}$ |
| | #Training epochs | 5 | 8 | 5 |
| Finetune-Raw | Learning rate | $5 \times 10^{-5}$ | $1 \times 10^{-4}$ | $5 \times 10^{-5}$ |
| | #Training epochs | 8 | 16 | 8 |

We also perform model-specific tuning of the learning rate for the experiments using Gemma 2 and Qwen 3 as the backbone models, while keeping the number of training epochs consistent with the Llama 3 configurations. The resulting hyperparameter sets are detailed in Table 6. All LIFT experiments are conducted using a cluster of 8 × 80GB A100 or 90GB H20 GPUs. Given that the model instance can fit within a single GPU, we adopt data parallelism to accelerate the fine-tuning process.

*Table 6.* The hyperparameters employed in the experiments of LIFT based on Gemma 2 and Qwen 3.

| Backbone model | Learning rate | #Train epochs |
|----------------|---------------|---------------|
| Gemma 2 | $1.0 \times 10^{-3}$ | 5 |
| Qwen 3 | $5 \times 10^{-5}$ | 5 |

## B. Synthetic Task Generator Prompts

We employ Qwen2.5-72B-Instruct as the generator to generate synthetic tasks in the experiments. The generator is deployed on a cluster of 8 × H20 GPUs using the vLLM inference engine to ensure high throughput and asynchronous generation. To promote context consistency of the synthetic tasks, especially the issues caused by incomplete information (e.g., pronouns with unresolved references), we provide each sentence to the generator along with a short preceding paragraph as context. This ensures that the generator can fully interpret the sentence, extract the relevant information, and represent it in the form of question–answer pairs. Moreover, to improve diversity, the 5 or 10 QA pairs corresponding to a single sentence are generated in a single interaction turn. Consequently, the generator has access to the previously generated QA pairs when generating a new one, allowing it to avoid redundancy. To align the synthetic tasks with the specific requirements of each benchmark, we employ few-shot prompting. The prompts for SQuAD, NIAH, and LooGLE are presented as follows, respectively:

**The prompt for SQuAD** We select two set of context-based QA pairs from the training split of SQuAD as the few-shot examples.

---
**System:**
# Instruction
You are given a paragraph extracted from an article. Please read it and generate at most {*num_questions*} different questions based on the content of the **last part** of the paragraph. The questions should be diverse in both their

---

form and the content they inquire about.
# Examples
Please follow the examples below to generate question-answer pairs from the given paragraph.
Example 1:
    Paragraph: After Hurricane Katrina in 2005, Beyoncé and Rowland founded the Survivor Foundation to provide transitional housing for victims in the Houston area, to which Beyoncé contributed an initial $250,000. The foundation has since expanded to work with other charities in the city, and also provided relief following Hurricane Ike three years later.
    Question1: How much did Beyonce initially contribute to the foundation?
    Answer1: $250,000
    Question2: How has this foundation changed in recent years?
    Answer2: expanded to work with other charities
    Question3: What foundation did Beyoncé start after Hurricane Katrina?
    Answer3: Survivor Foundation
    Question4: What other hurricane did Beyoncé's foundation help with?
    Answer4: Hurricane Ike
Example 2:
    Paragraph: With a total area of 147,040 square miles (380,800 km2), Montana is slightly larger than Japan. It is the fourth largest state in the United States after Alaska, Texas, and California; the largest landlocked U.S. state; and the 56th largest national state/province subdivision in the world. To the north, Montana shares a 545-mile (877 km) border with three Canadian provinces: British Columbia, Alberta, and Saskatchewan, the only state to do so. It borders North Dakota and South Dakota to the east, Wyoming to the south and Idaho to the west and southwest.
    Question1: What state does Montana border to the south?
    Answer1: Wyoming
    Question2: What state does it border to the west?
    Answer2: Idaho
    Question3: Where are most of the states mountain ranges?
    Answer3: western half of the state
    Question4: How much of the state is prarie?
    Answer4: About 60 percent
# Output format
Output in the JSON format. DO NOT output anything else.
{
    "qa_list": [
        {"question": ..., "answer": ...},
        {"question": ..., "answer": ...},
        ...
    ]
}
**User:**
The paragraph:
{*paragraph*}

The last part of the paragraph:
{*the target sentence*}

Generate {*num_questions*} different questions based on the content of the last part of the paragraph.

**The prompt for NIAH** We manually construct a NIAH-style QA pair as an example.

**System:**
# Instruction

You are given a paragraph extracted from an article. Please read it and generate at most {*num_questions*} different questions based on the content of the **last part** of the paragraph. The questions should be diverse in both their form and the content they inquire about.
# Example:
The last part of the paragraph: The perfect way to start a Sunday in Portland is to grab a flat white from a neighborhood café, browse the shelves at a cozy used bookstore, and then take a slow walk through Forest Park.
Question: What is the perfect way to start a Sunday in Portland?
Answer: The perfect way to start a Sunday in Portland is to grab a flat white from a neighborhood café, browse the shelves at a cozy used bookstore, and then take a slow walk through Forest Park.
# Output format
Output in the JSON format. DO NOT output anything else.
{
   "qa_list": [
      {"question": ..., "answer": ...},
      {"question": ..., "answer": ...},
      ...
   ]
}
**User:**
The paragraph:
{*paragraph*}

The last part of the paragraph:
{*the target sentence*}

Generate {*num_questions*} different questions based on the content of the last part of the paragraph.

**Prompt for LooGLE** Due to the diversity of the test tasks in LooGLE, we do not provide any example to the generator to encourage it to cover as much information as possible.

**System:**
# Instruction
You are given a paragraph extracted from an article. Please read it and generate at most {*num_questions*} different questions based on the content of the **last part** of the paragraph. The questions should be diverse in both their form and the content they inquire about.
# Output format
Output in the JSON format. DO NOT output anything else.
{
   "qa_list": [
      {"question": ..., "answer": ...},
      {"question": ..., "answer": ...},
      ...
   ]
}
**User:**
The paragraph:
{*paragraph*}

The last part of the paragraph:
{*the target sentence*}

Generate {*num_questions*} different questions based on the content of the last part of the paragraph.

## C. GPT4-Score Evaluation

We utilize GPT-4 to evaluate the correctness of the responses of LLMs based on the questions and the ground-truth answers on SQuAD and LooGLE, which has been proven to be consistent with human evaluation. The prompt is as follows:

> Given one question, there is a groundtruth and a predict_answer. Please decide whether they are the same or not in semantic. Please only output 'True' or 'False' .
> Question: {*Question*}
> groundtruth = {*Ground-truth answer*}
> predict_answer = {*LLM response*}

## D. Comparison against LLoCO

LLoCO (Tan et al., 2024) is a recent method designed to accelerate long-context processing by compressing long inputs into dense latent tokens. It utilizes AutoCompressor (Chevalier et al., 2023) to compresses long documents into summary embeddings, which are subsequently used as soft prompts during inference. Before evaluating on a downstream task, to improve the model's ability to interpret the summary tokens, LLoCO fine-tunes the model (Llama-2-7b-instruct) on the training split of the downstream task. Given that LooGLE resembles QuALITY (Pang et al., 2022) in the tasks – both focusing on long-context-based question-answering – we follow the process of evaluating LLoCO on QuALITY in its original paper to evaluate LLoCO on LooGLE. As LooGLE does not provide a training split, we randomly partition it into a training and test split, ensuring that the training set size is comparable to that of QuALITY training split.

Moreover, as our main experiments are based on Llama 3, Gemma 2, and Qwen 3, which have stronger foundational capabilities than Llama 2, we conduct an additional experiments of LIFT based on Llama 2 for a fair comparison. The configuration is the same as the standard one used in LIFT with 10QA, except that it's based on Llama 2 and is evaluated on the test split mentioned before. The results are illustrated in Table 7.

*Table 7.* Comparison between LIFT and LLoCO on LooGLE.

| Method | ShortQA | LongQA | Comprehension & Reasoning | Multiple info retrieval | Computation | Timeline reorder |
|--------|---------|--------|---------------------------|-------------------------|-------------|------------------|
| LLoCO | 21.04 | 21.92 | 29.82 | 17.10 | 0.00 | 23.71 |
| LIFT | 54.65 | 29.26 | 38.53 | 24.35 | 5.68 | 28.87 |

## E. Baseline Reproduction Details

### E.1. MemoryLLM

Generally, we adopt the official checkpoint memoryllm-8b-chat and the same method to process the documents in LooGLE and LongBench as the official implementation of MemoryLLM. For LooGLE, we split the tokenized document into consecutive segments of length of 512 tokens and inject the segments sequentially into the model memory, and prompt the model to answer the question without providing the document in the context. The prompt is as follows:

> Please answer the following question: {*Question*}

The context is injected into the model memory the same as the process of the LooGLE documents and the model respond to the prompt without access to the context.

### E.2. LlamaIndex

We adopt bge-small-en-v1.5 as the embedding model and Llama-3-8B-Instruct as the generator for LlamaIndex. Since each task of LooGLE and LongBench are based on a single context, we provide the context (without prompts) to LlamaIndex as a single document, and evaluate its ability to answer questions given only the prompt.

### E.3. TempLoRA

TempLoRA is a method designed for long generation tasks. It iteratively internalizes context – either user-provided or model-genereated – into model parameters. To evaluate TempLoRA on LooGLE, we follow the evaluation pipeline established for PG19 (Rae et al., 2019), a long book language modeling benchmark. Specifically, we append a question to the corresponding context, transmitting the task into a continue-writing task, thereby aligning the LooGLE tasks with the original setting of TempLoRA.

### E.4. LLMLingua

LLMLingua is a prompt compression approach, which compresses long context using a LLM (the compressor) to fit it in the context window of the target LLM (for Llama-3-8B-Instruct, the target length is 8192). We adopt the NousResearch/Llama-2-7b-hf checkpoint as the compressor, which is recommended by the authors. The compressed context is subsequentially provided to Llama-3-8B-Instruct to answer questions. Notice that we do not compress the questions together with the contexts, as the questions are much shorter than the contexts.

## F. Examples of the Synthetic Tasks

*Table 8.* Examples of the synthetic tasks generated for the needle "The best thing to do in San Francisco is eat a sandwich and sit in Dolores Park on a sunny day.". Note that the generator also produces synthetic tasks for distracting contexts as it does not know the needle at test time.

| Example 1 | |
| --- | --- |
| Question | What is recommended as the best activity to in San Francisco? |
| Answer | The best thing to do in San Francisco is eat a sandwich and sit in Dolores Park on a sunny day. |
| Example 2 | |
| Question | What is the best thing to do in San Francisco according to the sentence? |
| Answer | The best thing to do in San Francisco is eat a sandwich and sit in Dolores Park on a sunny day. |

The demonstrations of the synthetic tasks on NIAH are illustrated in Table 8. The demonstrations of the synthetic tasks on LooGLE are listed as follows:

> **Q1:** Where was Picardo born?
> **A1:** Jerez de la Frontera, in the Province of Cádiz in Andalucía, Spain
>
> **Q2:** In which province is Jerez de la Frontera located?
> **A2:** Province of Cádiz
>
> **Q3:** What region of Spain is Jerez de la Frontera in?
> **A3:** Andalucía
>
> **Q4:** On what date was Picardo born?
> **A4:** 18 June 1919
>
> **Q5:** How old would Picardo be if he were alive today?
> **A5:** To calculate Picardo's age if he were alive today, we need to know the current year and subtract his birth year

from it. Picardo was born in 1919. If the current year is 2023, then:

2023 - 1919 = 104

Picardo would be 104 years old if he were alive today.

**Q6:** What was the full name of Picardo's father?
**A6:** Alvaro Picardo de Celis

**Q7:** What was the family name of Picardo's mother?
**A7:** Castellón

**Q8:** How might Picardo's full name have been influenced by his parents' names?
**A8:** Picardo's full name likely included his father's surname, Picardo, and his mother's surname, Castellón, following the Spanish naming tradition. Thus, his full name might have been something like Francisco Picardo Castellón.

**Q9:** Is there any indication that Picardo had a middle name or additional surnames based on the information provided?
**A9:** No, there is no indication that Picardo had a middle name or additional surnames based on the information provided.

**Q10:** In Spanish naming customs, how would Picardo's children typically combine their surnames?
**A10:** In Spanish naming customs, Picardo's children would typically combine their surnames by using both Picardo (from their father) and Castellón (from their mother). For example, if Picardo had a son named Juan, he would be named Juan Picardo Castellón.

## G. Evaluation of the Fundamental Capabilities of LIFTed Models

While fine-tuning a pretrained model on a downstream task significantly improves its performance on that task, it may suffer from a severe degradation of other capabilities (a.k.a. catastrophic forgetting), especially when the model overfits to the downstream task. To investigate whether LIFT compromises the fundamental capabilities of the backbone model, we evaluate the LIFTed models on threee popular benchmarks: MMLU (0-shot, w/o CoT), GSM8K (8-shot, w/o CoT), and ARC Challenge (0-shot), focusing on the fundamental capabilities of LLMs.

We employ Language Model Evaluation Harness (Gao et al., 2024) to evaluate the pretrained model and 10 LIFTed models, which are independently fine-tuned on the first 10 documents in LooGLE ShortQA respectively. We report the average metrics of the 10 LIFTed models on the benchmarks. The results are presented in Table 9. We observe that the LIFTed models even slightly outperforms the pretrained model. While LIFT shows marginal gains on these benchmarks, we characterize these results as maintaining the capabilities rather than a fundamental improvement in general intelligence. The marginal gains observed are attributed to improved task alignment and enhanced model calibration during the fine-tuning process. The reason why LIFT does not significantly harm the model's fundamental capabilities is that fine-tuning on QA pairs eliminates the need for the model to memorize excessive lexical details. Unlike fine-tuning on raw context, it does not require overfitting to the synthetic tasks to understand contextual knowledge.

*Table 9.* Compare the LIFTed models against the pretrained models on MMLU, GSM8K, and ARC Challenge. The backbone model is Llama-3-8B-Instruct.

| Model | MMLU (avg. score) | GSM8K (flexible extract) | ARC Challenge (accuracy) |
|---|---|---|---|
| Pretrained | 58.13 | 50.80 | 48.63 |
| LIFTed | 62.26 | 52.01 | 52.39 |

## H. Additional Experiments on the Variants of LIFT

In the standard LIFT setting, the LLM is fine-tuned exclusively on synthetic tasks and it is not provided with context during inference. We conduct additional experiments by designing three variants of LIFT: first, the model is fine-tuned on both

synthetic tasks and raw context segments, denoted as **+segmented LM**; second, during inference, the context is provided but is truncated to fit the 8K context window, denoted as **+truncated ICL**; third, this variant combines +segmented LM and +truncated ICL, denoted as **+both**. The performance of LIFT and its variants on LooGLE using Llama-3-8B-Instruct as the foundation model is presented in Table 10.

*Table 10.* Performance of LIFT and its variants on LooGLE based on the Llama-3-8B-Instruct.

| Methods | ShortQA | LongQA | Comprehension & Reasoning | Multiple info retrieval | Computation | Timeline reorder |
|---|---|---|---|---|---|---|
| LIFT with 5QA | 45.67 | 26.79 | 29.80 | 21.58 | 14.00 | 36.28 |
| + segmented LM | 44.08 | 26.61 | 27.83 | 20.79 | 14.00 | 40.47 |
| + truncated ICL | 49.31 | 27.52 | 31.28 | 22.37 | 10.00 | 37.67 |
| + both | 50.28 | 26.70 | 29.31 | 23.42 | 11.00 | 34.88 |
| LIFT with 10QA | 52.69 | 27.25 | 27.83 | 22.63 | 16.00 | 39.53 |
| + segmented LM | 54.07 | 26.70 | 29.56 | 22.37 | 15.00 | 34.42 |
| + truncated ICL | 56.43 | 28.52 | 28.82 | 23.68 | 14.00 | 43.26 |
| + both | 55.10 | 24.34 | 26.11 | 20.53 | 11.00 | 33.95 |

Generally, integrating LIFT with truncated ICL provides the model with expanded information sources during inference. As anticipated, its performance is slightly better than the standard LIFT. However, we emphasize that LIFT represents a zero-to-one shift, enabling context-free question-answering, whereas truncated ICL yields only marginal gains. This suggests that LIFT successfully internalizes the vast majority of contextual knowledge. Moreover, we observe that segmented language modeling offers negligible benefits and often degrades performance. This is likely because fine-tuning on raw context segments requires the model to memorize all the tokens, which may compromise the model's general capabilities.

## I. Analysis of the Quality of Synthetic Data

Since LIFT relies on synthetic data to memorize and understand context, it's necessary to inspect the quality of the generated synthetic data. We examine both the format and content diversity of the synthetic tasks generated by Qwen2.5-72B for LooGLE benchmark. To assess format diversity, we prompt Qwen2.5-72B to rewrite all QA pairs into "what-is" style; to assess content diversity, we categorize the synthetic tasks into "event", "location", "person", "time", and "other" using Qwen2.5-72B and fine-tune the model on each category separately. The proportions of the categories are shown in Figure 7 and the evaluation results are illustrated in Table 11. The results indicate that diversity matters, but the generator is capable of producing diverse QA pairs.

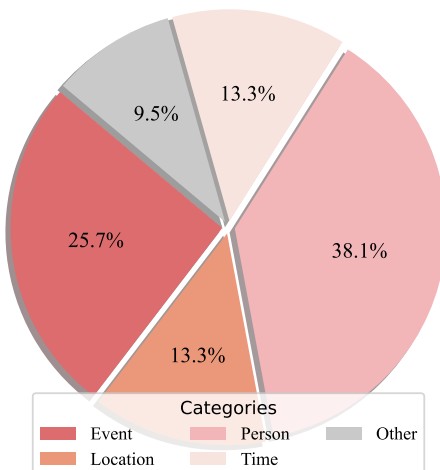

*Figure 7.* Distribution of Synthetic QA Tasks by Content Category.

The following are the examples of each category:

*Table 11.* Performance on LooGLE of Llama-3-8B-Instruct fine-tuned on synthetic tasks formatted as "what-is" questions, or on synthetic tasks from only one of the five task categories.

|  | What-is | Event | Location | Person | Time | Other |
|---|---|---|---|---|---|---|
| ShortQA | 40.11 | 33.84 | 25.70 | 40.54 | 23.34 | 23.34 |
| LongQA | 13.78 | 17.82 | 16.37 | 17.91 | 21.44 | 13.93 |

---

**Event**
**Q**: What unique item did Picardo design for the fashion brand Loewe in 1959?
**A**: He designed a pack of playing cards (baraja de naipes) for their exclusive advertising use.

**Location**
**Q**: Where was José Luis Picardo born?
**A**: He was born in Jerez de la Frontera, located in the Province of Cádiz, Andalucía, Spain.

**Person**
**Q**: Whose architectural studio did Picardo join to avoid evacuation during the Spanish Civil War?
**A**: He joined the studio of architect and professor Luis Moya Blanco.

**Time**
**Q**: When did the Parador de Guadalupe officially open to the public?
**A**: The hotel opened on December 11, 1965.

**Other**
**Q**: What was the significance of the "Manifiesto de la Alhambra" that Picardo signed in 1952?
**A**: It is considered one of the most remarkable texts in 20th-century Spanish architectural historiography and sought inspiration from the Alhambra for a distinctively Spanish form of modern architecture.

---

## J. Using the Target Model as Synthetic Data Generator

In our paper, we use Qwen2.5-72B as the generator, which is capable of producing diverse and high-quality QA pairs. However, a large generator is not necessary to guarantee the quality of the synthetic data. Moreover, LIFT's advantage stems from its understanding of context, rather than from knowledge distilled from a strong generator. To verify this, we conducted an additional experiment in which the target model itself (Meta-Llama-3-8B-Instruct) serves as the generator. The results are shown in Table 12, which suggest that an 8B model is already sufficient to generate useful synthetic tasks, and that the effectiveness of LIFT does not critically depend on a strong generator.

*Table 12.* Comparison of LIFT with the target model (Llama-3-8B-Instruct) itself as the generator and the ICL baseline.

| Method | ShortQA | LongQA | Comprehension & Reasoning | Multiple info retrieval | Computation | Timeline reorder |
|---|---|---|---|---|---|---|
| ICL | 44.49 | 15.44 | 25.37 | 15.26 | 5.00 | 1.86 |
| LIFT | 48.49 | 26.93 | 34.14 | 21.49 | 15.00 | 28.32 |

## K. Extending LIFT to Diverse Task Formats

To evaluate the generalizability of LIFT beyond document-based question answering (e.g., SQuAD and LooGLE), we assess its performance on document summarization (GovReport, Huang et al. (2021)) and a specialized in-context skill acquisition scenario derived from GSM8K. This setup tests whether LIFT can internalize complex reasoning patterns and structural constraints into model parameters, rather than merely retrieving facts.

In the GSM8K skill acquisition task, the input context consists of 10 solved examples. The model must learn the underlying reasoning logic from these examples to solve a separate, unseen test case. For both benchmarks, we compare LIFT

against a truncated in-context learning baseline. We utilize Qwen2.5-72B-Instruct as the synthetic task generator and Llama-3-8B-Instruct as the target model.

For LIFT, we design task-specific strategies to generate synthetic tasks:

- **GovReport**: The generator is prompted to produce 3 summarizations of their granularity from coarse to fine every 20 sentences. The target model is fine-tuned on the summarizations.

- **GSM8K**: For each of the 10 examples, the generator extracts the reasoning steps and rewrites them while keeping the meaning. Moreover, it synthesizes 10 more examples sharing a similar solving strategy with the example.

As shown in Table 13, LIFT consistently outperforms the ICL baseline. Notably, the significant gain on GSM8K (+7.8%) demonstrates that LIFT's synthetic task generation strategy is superior to in-context learning, allowing the model to learn from contexts better.

*Table 13.* Performance comparison between LIFT and the truncated ICL baseline on GovReport and the in-context skill-acquisition task (GSM8K).

| Method | GovReport | GSM8K |
|--------|-----------|-------|
| ICL    | 29.1      | 57.6  |
| LIFT   | 30.9      | 65.4  |

