# OpenReview forum: "LIFT: A Novel Framework for Enhancing Long-Context Understanding of LLMs via Long Input Fine-Tuning"
_ICML.cc/2026/Conference — ICML 2026 regular_

### Official Review · Reviewer_zHHq · 2026-03-07

**Soundness:** 2
**Presentation:** 2
**Significance:** 2
**Originality:** 2
**Overall Recommendation:** 4
**Confidence:** 4

**Summary:**

This paper presents LIFT, a framework that attempts to improve the long-context processing capability of small-sized LLMs through test-time fine-tuning with synthetically generated QA pairs. The core idea is that, given a long input document at inference time, the system first uses an external LLM (or the model itself) to generate synthetic QA tasks grounded in the input, then fine-tunes the model's parameters on these synthetic tasks before answering the actual user query. The authors argue that this procedure effectively "teaches" the model to internalize the long context into its parameters rather than relying solely on in-context attention over long token sequences. The framework includes pipeline-level optimizations such as parallel generation and asynchronous scheduling to reduce the overhead introduced by test-time fine-tuning.

**Compliance With Llm Reviewing Policy:**

Affirmed.

**Final Justification:**

The authors have addressed all my concerns. I've raised my score.

**Key Questions For Authors:**

1. This paper frames LIFT as an alternative to RAG, yet the fundamental operation of LIFT (generate QA pairs from the document, then fine-tune) is arguably a more expensive and less flexible variant of RAG. RAG retrieves relevant chunks at inference time with negligible latency, whereas LIFT requires generating synthetic data and performing gradient updates per input. The paper does not provide a convincing argument for why encoding knowledge into parameters via fine-tuning is fundamentally preferable to encoding it via in-context retrieval, especially when the reported performance gains over RAG are modest on several benchmarks.
2. The method uses an external, presumably larger LLM (e.g., GPT-series or a larger open-source model) to generate the synthetic QA pairs that are then used to fine-tune the smaller target model. This introduces a significant hidden cost: the compute and API expense of running the external model over the entire long document is not accounted for in the efficiency analysis. If one already has access to a powerful LLM capable of reading and generating QA over the full long context, the practical value of then fine-tuning a smaller model becomes less clear.
3. The generated QA pairs appear to be mostly extractive or short-answer in nature (e.g., "What is X mentioned in paragraph Y?"). This biases the fine-tuning toward fact retrieval rather than deeper comprehension, summarization, or reasoning. The paper itself acknowledges that LIFT struggles with tasks requiring multi-hop reasoning or information integration, but this is a fundamental limitation of the methodology, not merely a boundary condition. It raises the question of what "long-context understanding" truly means in this framework versus simple information memorization.
4. SQuAD is an extractive QA benchmark, NIAH is a synthetic retrieval test, and LooGLE, while more diverse, still predominantly tests localized fact recall. There is no evaluation on tasks that genuinely require holistic long-document understanding, such as long-document summarization (e.g., GovReport, BookSum), multi-document QA (e.g., MultiDoc2Dial), or narrative understanding benchmarks (e.g., QuALITY, NarrativeQA). Without these, the claim of improving "long-context understanding" is overstated relative to what is actually demonstrated.
5. The paper ablates the number of synthetic QA pairs but does not systematically study the quality of these pairs. What happens when the external LLM generates incorrect, ambiguous, or trivially redundant QA pairs? Is there any filtering mechanism? How sensitive is LIFT to the quality of the QA generator? Given that the entire framework hinges on the quality of synthetic data, the lack of a robustness analysis here is a notable gap.
6. This paper discusses efficiency optimizations but does not clearly characterize how the total wall-clock cost scales as document length increases (e.g., from 16K to 64K to 128K tokens). As documents grow longer, both the number of synthetic QA pairs and the fine-tuning steps presumably need to increase. Does the TTFT grow linearly, super-linearly, or sub-linearly with input length? A clear scaling curve would be essential to assess practical viability, but is not provided in sufficient detail.

**Limitations:**

Please refer to the key question.

**Strengths And Weaknesses:**

1. The paper targets a real and practically relevant problem, which enables small LLMs with limited context windows (e.g., 4K or 8K tokens) to handle long inputs that far exceed their native capacity. This is a meaningful direction, as deploying large-context models is often infeasible due to resource constraints.
2. The paper does not treat the fine-tuning pipeline as a black box but explicitly addresses efficiency concerns through parallel synthetic data generation and asynchronous scheduling. The inclusion of a TTFT analysis provides a practical lens on the deployment overhead, which is often missing in similar work.

---

> ### Author Rebuttal · Authors · 2026-03-31
>
> Thank you for the detailed comments. We address each concern below.
>
> **1. Why is encoding knowledge into parameters via LIFT preferable to in-context retrieval?**
>
> (a) **LIFT is necessary when a long input defines a new document-specific skill to internalize, rather than retrievable facts.** In such cases, RAG is limited because downstream queries may differ greatly from the document's wording. By contrast, LIFT produces diverse, downstream-tailored examples, enabling the model to internalize the skill rather than rely on brittle lexical retrieval.
>
> Such applications are common in practice, including:
> - **Enterprise compliance review**: queries often describe detailed scenarios, while the rules are abstract and hard for RAG to retrieve.
> - **Complex logical reasoning**: the context consists of the model's exploratory trials, from which training can internalize abstract heuristics, but these learned heuristics are difficult for RAG to retrieve.
>
> We also add an experiment showing that LIFT supports skill learning beyond fact memorization; please see Point 3.
>
> (b) **LIFT substantially reduces token usage in multi-turn interactions.** As RAG repeatedly injects retrieved spans, the history quickly reaches the context limit and must be truncated, often hurting performance. By contrast, LIFTed models answer queries without context, saving most tokens. To quantify this advantage, we use a document from LooGLE as the context and ask 10 document-grounded questions sequentially. Using Qwen3-8B (40960-token window), the token costs are:
>
> |Turn|1|2|3|4|5|6|7|8|9|10|
> |-|-|-|-|-|-|-|-|-|-|-|
> |RAG|1241|2564|4846|7153|9489|11771|14071|16292|18467|20784|
> |LIFT|42|69|97|123|161|197|225|255|280|312|
>
> **2. What is the advantage of fine-tuning a small model if a large one is available?**
>
> LIFT's key advantage is **avoiding full long-context processing**. During synthesis and training, the context is split into sentences instead of being fed in full; at inference, no context is needed. A large long-context model, by contrast, must repeatedly process the full context. We also show that **LIFT remains effective when taking the small model itself as the generator**; please see Point 3 of our response to Reviewer 8YV6.
>
> **3. Does LIFT merely improve the model’s factual memorization or it improve the comprehension of the document?**
>
> We emphasize that LIFT provides a zero-to-one improvement, requiring no inference-time context. Despite this constraint, it outperforms all baselines on LooGLE ShortQA and LongQA, which test factual recall and long-context reasoning, respectively. We consider synthesizing long-dependency tasks as a future direction to further improve LIFT's performance. Moreover, we conducted additional experiments on **GovReport** (summarization) and **QuALITY** (narrative understanding) to demonstrate LIFT's advantage across a broader range of tasks. We also reformulated **GSM8K** as a new-skill-learning task. The ICL baseline applies a few-shot learning strategy with 10 examples, whereas LIFT reads the same 10 examples, synthesizes solving strategies and additional examples, and fine-tunes the model accordingly. The results are as follows:
>
> ||GovReport|QuALITY|GSM8K|
> |-|:-:|:-:|:-:|
> |LIFT|30.93|61.22|65.4|
> |ICL|29.10|45.13|57.6|
>
> **4. What if the generator is noisy? Is there any method to improve the quality of the synthetic tasks?**
>
> We agree that generator quality matters. In fact, we **carefully designed the synthesis pipeline to improve data quality** -- to reduce missing-context errors, we provide the target sentence together with a short preceding span; to reduce redundancy, we generate the multiple QA pairs for the same sentence in a single pass, allowing the generator to avoid previously produced QA pairs. Moreover, **the synthesis task itself is relatively simple**, and in the additional experiments we find that LIFT remains effective even when using the small model itself as the generator. Finally, **LIFT is empirically robust to imperfect synthetic data**: on LooGLE, the accuracy of the generated synthetic tasks estimtated by GPT-4 is 75.82%, yet LIFT still outperforms all baselines.
>
> **5. Does the TTFT grow linearly, super-linearly, or sub-linearly with input length?**
>
> Theoretically, assuming that the number of sentences grows linearly with the context length and the length of each QA pair remains constant, then the training cost also grows linearly with the context length. Additionally, since the LIFTed model requires no context during inference, the inference cost remains constant. Therefore, the overal complexity is linear. To validate this, we extend the TTFT evaluation to inputs of up to 16K tokens. The results are shown below:
>
> |Input Length|2K|4K|6K|8K|10K|12K|14K|16K|
> |-|-|-|-|-|-|-|-|-|
> |TTFT|4.47|5.99|7.53|9.54|11.20|13.08|15.61|17.20|
>
> The fitting curve is *TTFT≈2.246+0.926×Length(K)* with *R²≈0.996*, which is a strong evidence of linear scaling.

---

> > ### Author Rebuttal · Reviewer_zHHq · 2026-04-01
> >
> > The rebuttal clarifies my concerns. I will raise my score.

---

> > > ### Author Response · Authors · 2026-04-02
> > >
> > > We are very grateful for your detailed review and thoughtful comments. We are glad that our revisions addressed your concerns, and we sincerely thank you for updating the score. Your acknowledgment of the improvements and the paper's contributions is appreciated.

---

### Official Review · Reviewer_8YV6 · 2026-03-11

**Soundness:** 3
**Presentation:** 3
**Significance:** 3
**Originality:** 3
**Overall Recommendation:** 4
**Confidence:** 3

**Summary:**

The paper proposes LIFT, a framework for long-context modeling  that dynamically adapts its parameters given the input. Unlike previous works, LIFT internalizes the long document in its parameters instead of mere memorization. Experiments on different datasets show that this approach improves document-based QA and long-context information retrieval over several baselines.

**Compliance With Llm Reviewing Policy:**

Affirmed.

**Final Justification:**

The rebuttal clarifies my concerns. I confirm my original score.

**Key Questions For Authors:**

See **Weakness**

**Limitations:**

yes

**Strengths And Weaknesses:**

**Strengths**

The paper presents an interesting and unconventional perspective on long-context modeling: storing document knowledge in parameters rather than in the context window.
The motivation for using synthetic QA instead of raw-text fine-tuning is clear and supported by both qualitative examples and empirical results.
The proposed method is simple but effective, showing promising gains on the long-context benchmarks. It provides a totally different paradigm compared with previous works.

**Weakness**

1. The LIFT method requires synthetic task generation and fine-tuning for each new input, which may introduce practical challenges for deployment.
2. Moreover, it is better viewed as a document-specific test-time adaptation approach rather than a general long-context modeling solution.
3. The effect of the quality and coverage of the synthetic QA generator is also underexplored.

---

> ### Author Rebuttal · Authors · 2026-03-31
>
> We are grateful for your comments and respond to each of your points in turn.
>
> **1. The LIFT method requires synthetic task generation and fine-tuning for each new input, which may introduce practical challenges for deployment.**
>
> We agree that LIFT introduces a document-specific adaptation cost for each new input. However, in many practical cases, a long document supports multiple downstream queries. In such cases, LIFT's finetune-once, answer-many advantage becomes substancial and can offset the deployment cost. After the one-time training stage, the model no longer requires the original long context at inference, avoiding the repeated cost of long-context processing for every query. Moreover, recent adapter-serving frameworks also provide a practical path for hosting document-specific LoRA adapters. Finally, our additional experiment (please refer to the 3rd point of our response) indicates that the generator and the target model can be the same, which further reduces the deployment cost by removing the need for a separate generator.
>
> **2. Moreover, it is better viewed as a document-specific test-time adaptation approach rather than a general long-context modeling solution.**
>
> We agree that LIFT is a form of test-time adaptation. However, we view it as a combination of test-time adaptation and long-context modeling, rather than only the former: **its goal is still to enable efficient handling of long inputs, but through in-parameter adaptation rather than extending the context window**. In particular, by processing the context sentence-by-sentence during synthesis and eliminating the need to keep the long input in context during inference, LIFT avoids the quadratic cost of long-context attention and substantially reduces decoding cost, which is often the dominant component in long-context processing. We highlight LIFT’s efficiency improvements on long-context modeling in Figure 6 of our paper.
>
> **3. The effect of the quality and coverage of the synthetic QA generator is also underexplored.**
>
> In our paper, we use Qwen2.5-72B as the generator, which is capable of producing diverse and high-quality QA pairs. However, a large generator is not necessary to guarantee the quality of the synthetic data. To verify this, we conducted an additional experiment in which the target model itself (Meta-Llama-3-8B-Instruct) serves as the generator. The results are shown below:
>
> |Method|ShortQA|LongQA|Comprehension and reasoning|Multiple information retrieval|Computation|Timeline reorder|
> |-|-|-|-|-|-|-|
> |ICL|44.49|15.44|25.37|15.26|5.00|1.86|
> |LIFT (Qwen2.5 72B)|52.69|27.25|27.83|22.63|16.00|39.53|
> |LIFT (Llama3 8B)|48.49|26.93|34.14|21.49|15.00|28.32|
>
> Even when using the much smaller Meta-Llama-3-8B-Instruct as the generator, LIFT still consistently outperforms the baseline by a large margin. This shows that **an 8B model is already sufficient to generate useful synthetic tasks**, and that the effectiveness of LIFT does not critically depend on a particularly strong generator.

---

> > ### Author Rebuttal · Reviewer_8YV6 · 2026-04-01
> >
> > The rebuttal clarifies my concerns. I confirm my original score.

---

> > > ### Author Response · Authors · 2026-04-02
> > >
> > > Thank you for your careful review and thoughtful feedback. We are glad that our revisions clarified your concerns and appreciate the time you put to evaluating our work.

---

### Official Review · Reviewer_42Ct · 2026-03-12

**Soundness:** 3
**Presentation:** 4
**Significance:** 3
**Originality:** 3
**Overall Recommendation:** 4
**Confidence:** 4

**Summary:**

This paper proposes Long-Input Fine-Tuning (LIFT), a novel framework designed to enhance the long-context understanding of large language models. Unlike conventional approaches that extend context windows, LIFT stores and absorbs the long input in parameters by fine-tuning the models on carefully designed LLM-generated synthetic tasks. To mitigate the computational overhead introduced by fine-tuning, the authors propose a highly optimized asynchronous pipeline that significantly reduces Time to First Token (TTFT). Experimental results on several benchmarks show the improvement of LIFT in performance and efficiency above competing method.

**Compliance With Llm Reviewing Policy:**

Affirmed.

**Final Justification:**

The rebuttal clarifies my concerns, so I confirm my original score.

**Key Questions For Authors:**

See Weaknesses part.

**Limitations:**

yes

**Strengths And Weaknesses:**

**Strengths**
1. This paper provides a strong and well-validated motivation for using synthetic QA pairs over raw text for fine-tuning. The pilot experiment on SQuAD, including a manual analysis of failure modes (Figure 2, Table 2), convincingly demonstrates the limitations of superficial pattern matching in Finetune-Raw and highlights the superior comprehension achieved by Finetune-QA.
2. The method is evaluated on multiple benchmarks (SQuAD, NIAH, LooGLE) with diverse task requirements (short QA, long QA, information retrieval, reasoning) and the results show the superiority of LIFT over competing method. The inclusion of multiple backbone models (Llama-3, Gemma 2, Qwen 3) in Section 4.5 effectively demonstrates the generalizability of LIFT.
3. This paper provides a valuable analysis of the method's efficiency in section 4.6, including the benefits of the asynchronous pipeline and the total time comparison against ICL. Figure 6 clearly illustrates the practical advantage of LIFT for generating longer outputs.

**Major Weaknesses**

1. The paper relies heavily on a powerful generator (Qwen-2.5-72B-Instruct) to create synthetic tasks but lacks systematic analysis of how the quality, quantity, and diversity of synthetic data impact the final performance of LIFTed models. Additionally, there exists a substantial capacity gap between the generator (72B parameters) and the target models (<10B parameters). The paper does not adequately address whether the observed performance gains primarily stem from the generator's strong information extraction capabilities rather than the LIFT framework itself. A critical missing baseline is directly feeding generator-produced answers to the target model without fine-tuning, which would help isolate the true contribution of the LIFT framework.
2. LIFT requires independent fine-tuning of the foundation model for each input instance, rendering it incompatible with batch inference which is a common technique for improving throughput in production environments. This instance-specific adaptation may limit the practical deployability of LIFT in high-throughput scenarios.
3. While LIFT demonstrates efficiency advantages over ICL when generating outputs exceeding 1K tokens, Figure 6(a) indicates significant overhead in Time to First Token (TTFT), particularly for shorter inputs. At zero output length (representing pure TTFT), the latency appears substantially higher than baseline methods, which could hinder adoption in latency-sensitive applications.

**Minor Weaknesses**
1. The paper lacks detailed description of the sentence segmentation strategy used to decompose input instances for synthetic QA generation.
2. The efficiency analysis would benefit from comprehensive TTFT comparisons against a broader range of baseline methods beyond ICL.

---

> ### Author Rebuttal · Authors · 2026-03-31
>
> We appreciate your detailed feedback and organize our response around the key points you raised.
>
> **1. Analysis of the impact of the quality, quantity, and diversity of synthetic data.**
> > The paper relies heavily on a powerful generator (Qwen-2.5-72B-Instruct) to create synthetic tasks but lacks systematic analysis of how the quality, quantity, and diversity of synthetic data impact the final performance of LIFTed models.
>
> (a) In our paper, we ablate the quantity of synthetic tasks(5 vs. 10 QA pairs per sentence). The results show that LooGLE ShortQA benefits more from a larger number of QA pairs, as more QA pairs cover a broader range of information.
>
> (b) We examine both the format and content diversity of the synthetic tasks. To assess format diversity, we prompt Qwen2.5-72B to rewrite all QA pairs into "what-is" style; to assess content diversity, we categorize the synthetic tasks into "event", "location", "person", "time", and "other" using Qwen2.5-72B and fine-tune the model on each category separately. The results indicate that **diversity matters, but the generator is capable of producing diverse QA pairs**. The evaluation results on LooGLE and the examples of the 5 categories are presented below:
>
> ||Original LIFT|"what-is"|event (27%)|location (14%)|person (40%)|time (14%)|other (10%)|
> |-|-|-|-|-|-|-|-|
> |ShortQA|52.69|40.11|33.84|25.70|40.54|23.34|23.34|
> |LongQA|27.25|13.78|17.82|16.37|17.91|21.44|13.93|
>
> - Event
>   - Question: What unique item did Picardo design for the fashion brand Loewe in 1959?
>   - Answer: He designed a pack of playing cards (baraja de naipes) for their exclusive advertising use.
> - Location
>   - Question: Where was José Luis Picardo born?
>   - Answer: He was born in Jerez de la Frontera, located in the Province of Cádiz, Andalucía, Spain.
> - Person
>   - Question: Whose architectural studio did Picardo join to avoid evacuation during the Spanish Civil War?
>   - Answer: He joined the studio of architect and professor Luis Moya Blanco.
> - Time
>   - Question: When did the Parador de Guadalupe officially open to the public?
>   - Answer: The hotel opened on December 11, 1965.
> - Other
>   - Question: What was the significance of the "Manifiesto de la Alhambra" that Picardo signed in 1952?
>   - Answer: It is considered one of the most remarkable texts in 20th-century Spanish architectural historiography and sought inspiration from the Alhambra for a distinctively Spanish form of modern architecture.
>
> **2. Whether does the performance arise from strong generator?**
> > The paper does not adequately address whether the observed performance gains primarily stem from the generator's strong information extraction capabilities rather than the LIFT framework itself.
>
> We conducted an additional experiment (please refer to the 3rd point of our response to Reviewer 8YV6) to show that LIFT works well without a powerful generator such as Qwen2.5-72B. When using the target small model, Meta-Llama-3-8B-Instruct, as the generator, LIFT still outperforms all baselines, demonstrating that LIFT's advantage arises from the pipeline design, rather than distillation from a stronger generator.
>
> **3. Applying LIFT to high-throughput or latency-sensitive scenarios.**
>
> > This instance-specific adaptation may limit the practical deployability of LIFT in high-throughput scenarios.
> > At zero output length (representing pure TTFT), the latency appears substantially higher than baseline methods, which could hinder adoption in latency-sensitive applications.
>
> We acknowledge that document-specific models pose deployment challenges when the number of documents is very large. However, in many applications, multiple queries are issued over the same document, in which case LIFT's finetune-once, answer-many advantage becomes substancial and can offset the deployment cost. Moreover, recent serving frameworks like vLLM support dynamic LoRA adapter serving (https://docs.vllm.ai/en/latest/features/lora/#serving-lora-adapters), making document-specific adapters practically deployable.
>
> While LIFT incurs a higher TTFT than ICL, we emphasize that TTFT reflects only the prefilling cost, whereas decoding often dominates the overall runtime. As shown in the efficiency analysis, LIFT achieves substantially faster decoding than ICL. Furthermore, in multi-query scenarios, the TTFT for subsequent queries after the initial training stage becomes nearly zero, since the model no longer requires the original context.

---

> > ### Author Rebuttal · Reviewer_42Ct · 2026-04-01
> >
> > My concerns have been addressed, so I will keep my score as it is.

---

> > > ### Author Response · Authors · 2026-04-02
> > >
> > > Thank you for your careful review and constructive suggestions. We are pleased that our revisions addressed your concerns. Your recognition of the paper's value is much appreciated.

---

### Official Review · Reviewer_UwgN · 2026-04-12

**Soundness:** 3
**Presentation:** 3
**Significance:** 3
**Originality:** 3
**Overall Recommendation:** 4
**Confidence:** 3

**Summary:**

The paper proposes LIFT, a test-time fintuning method for long-input QA. For each long input, the method automatically generates QA short samples based on the long input, and then the short QA samples are used for fintuning the target LLM, the finetuned LLM is finally used for testing time QA evaluation. The method achieves performance gain compared with based lines.

**Compliance With Llm Reviewing Policy:**

Affirmed.

**Final Justification:**

The idea is very interesting to me, and the method design is reasonable, so I vote for acceptance. However, I believe the method consumes much more time than ICL/RAG, so it cannot be really used in real cases, so I rate 4.

**Key Questions For Authors:**

I assume the time consumption of data generation and finetuning is a major block to stop such method into real usage. Does the authors see any hope in the future that we can apply this method but consuming comparative time as RAG or ICL?

**Strengths And Weaknesses:**

Strength:

1. Applying the finetuning idea with synthectic data generation to test-time long-input QA is interesting and new to me.

2. The desgin is reasonable. Since test dataset is QA, making finetuning data also QA is likely to improve the performance. Besides, the QA finetuning data is from a stronger model, so such distillation is likely to boost the performance.

Weakness:

1. How is the method LIFT compared with existing methods on time consumption? Since the method involved fientuning and data generation, I feel it consumes much more time than ICL or RAG.

2. The data generation process split long-context input to sentences as mentioned in Sec 3.2. Will this break information between sentences? For examples, we cannot have answer based on only partial sentences of the long input.

---

### Decision · Program_Chairs · 2026-04-30

**Decision:**

Accept (regular)

**Comment:**

**Summary:** This paper introduces LIFT, a test-time adaptation framework designed to internalize long documents into model parameters rather than relying on extended context windows. By generating synthetic QA pairs grounded in the input and fine-tuning the model via LoRA at inference time, the system enables short-context LLMs to answer queries without requiring the original text in the context. To mitigate the computational overhead of test-time training, the authors implemented an optimized asynchronous pipeline that maintains a Time to First Token (TTFT) of under 10 seconds for 8K-token inputs. Evaluation across SQuAD, NIAH, and LooGLE benchmarks demonstrates consistent improvements over ICL and RAG baselines.

**Summary of Reviews:** The reviewers reached a consensus with positive scores. They collectively noted that the core concept of shifting document knowledge from the context window into model parameters represents a highly interesting and unconventional perspective in long-context modeling. Reviewers highlighted the pilot experiments specifically designed to distinguish comprehension from superficial pattern matching as being particularly clear and well-motivated. However, the reviewers raised four primary concerns during the initial review phase. First, they questioned the practical deployment constraints, noting that instance-specific fine-tuning is incompatible with standard batch inference. Second, there was significant skepticism regarding the dependency on synthetic data, specifically whether performance gains were merely a result of distillation from a powerful 72B generator. Third, reviewers pointed out that the evaluation scope was largely restricted to extractive QA tasks, lacking evidence for holistic document understanding. Finally, the efficiency bottlenecks were scrutinized, particularly the high latency overhead for shorter inputs compared to traditional in-context learning.

**Assessment:** The authors successfully addressed these core criticisms through their rebuttal and supplementary experiments. Regarding deployment, they demonstrated that the "fine-tune once, answer many" advantage becomes substantial in multi-query scenarios and can be supported by modern serving frameworks. To address data dependency, new experiments confirmed that LIFT remains effective even when the target 8B model serves as its own generator. Furthermore, the authors expanded the evaluation scope by providing results on GovReport and QuALITY, demonstrating the framework's broader applicability. In terms of efficiency, the rebuttal provided a detailed scaling analysis showing that while initial TTFT is higher, LIFT achieves linear scaling and significantly faster decoding speeds than ICL for long-context tasks. While limitations in complex multi-hop reasoning persist, the paper offers a genuinely novel paradigm for parameter adaptation with clear practical utility.